# An Efficient Motion Planning Method with a Lazy Demonstration Graph for Repetitive Pick-and-Place

**DOI:** 10.3390/biomimetics7040210

**Published:** 2022-11-21

**Authors:** Guoyu Zuo, Mi Li, Jianjun Yu, Chun Wu, Gao Huang

**Affiliations:** 1Faculty of Information Technology, Beijing University of Technology, Beijing 100124, China; 2Beijing Key Laboratory of Computing Intelligence and Intelligent Systems, Beijing 100124, China; 3Beijing Advanced Innovation Center for Intelligent Robots and Systems, Beijing Institute of Technology, Beijing 100081, China

**Keywords:** manipulation planning, motion and path planning, learning sampling distribution, autonomous robot

## Abstract

Robotic systems frequently need to plan consecutive similar manipulation in some scenarios (e.g., pick-and-place tasks), leading to similar motion plans. Moreover, the workspace of a robot changes with the difference in operation actions, which affects subsequent tasks. Therefore, it is significant to reuse information from previous solutions for new motion planning instances to adapt to workplace changes. This paper proposes the Lazy Demonstration Graph (LDG) planner, a novel motion planner that exploits successful and high-quality planning cases as prior knowledge. In addition, a Gaussian Mixture Model (GMM) is established by learning the distribution of samples in the planning cases. Through the trained GMM, more samples are placed in a promising location related to the planning tasks for achieving the purpose of adaptive sampling. This adaptive sampling strategy is combined with the Lazy Probabilistic Roadmap (LazyPRM) algorithm; in the subsequent planning tasks, this paper uses the multi-query property of a road map to solve motion planning problems without planning from scratch. The lazy collision detection of the LazyPRM algorithm helps overcome changes in the workplace by searching candidate paths. Our method also improves the quality and success rate of the path planning of LazyPRM. Compared with other state-of-the-art motion planning algorithms, our method achieved better performance in the planning time and path quality. In the repetitive motion planning experiment of the manipulator for pick-and-place tasks, we designed two different experimental scenarios in the simulation environment. The physical experiments are also carried out in AUBO−i5 robot arm. Accordingly, the experimental results verified our method’s validity and robustness.

## 1. Introduction

Robot manipulators are widely used for performing continuous manipulation tasks such as parts assembly and material sorting in fields such as manufacturing [1]. In structured scenarios, robotic arms often only need to repeat the demonstration trajectory to perform repetitive tasks. However, in a semi-structured scene, the appearance of new obstacles will change the workspace, so that the denomination trajectory may collide with obstacles and affect the execution of operation tasks [2,3]. Furthermore, in some scenarios, the robotic arm often needs to perform multiple picking and placing tasks, the nature of repetitive pick-and-place tasks suggests that the solutions for all motion planning instances are similar to some extent [4]. In this case, we can collect successful planning cases during the offline phase, followed by reusing the knowledge online [5]. Importantly, two aspects should be considered in this case: (1) how to characterize and generalize the prior knowledge and use it to solve new motion planning problems; (2) how to ensure the stability of the motion planning process when the prior knowledge becomes invalid with changes of the environment.

In the past few decades, several methods have been proposed to solve motion planning problems. Among them, intelligent optimization algorithms, which iteratively solve trajectories that satisfy the task requirements, are popular. Tan et al. proposed a path planning method based on an improved ant colony algorithm [6], which improved the efficiency of planning by imperatively selecting the initial parameters of the ant colony algorithm through a particle swarm optimization algorithm. Evolutionary algorithms are used in robot trajectory planning to obtain a safe path that satisfies the kinematic constraints by iteratively optimizing the travel time and actuator effort [7]. However, intelligent optimization algorithms tend to fall into local extremes easily and converge slowly, which is determined by the nature of the optimization algorithm itself, while our method samples directly in the joint space, which is faster for planning and less likely to fall into local solutions.

Sampling-based motion planning (SBMP) algorithms can effectively solve motion planning problems without modeling the environment [8]; this type of method has improved robustness, therefore gradually becoming mainstream in this field. The Rapidly-exploring Random Tree (RRT) [9] uses a tree data structure to obtain connectivity information in the configuration space (C-space) of the robotic arm by random sampling. The Probabilistic Roadmap (PRM) [10] builds a roadmap for multi-query planning in C-space, so the roadmap can be reused for subsequent planning. However, in a semi-structured environment, before the roadmap is reused, it is necessary to perform collision detection on the nodes and edges of the roadmap again, otherwise the planned path may collide with obstacles. Some asymptotically optimal planning algorithms aim to minimize the cost function, such as RRT* and PRM* [11]. The path converges to the optimum with a sufficient number of samples. However, such algorithms often take a long time to converge because of the random sampling nature. Xu et al. proposed an RRTConnect algorithm based on a sparse expansion strategy and dead point saved strategy [12], which can effectively reduce the number of collision detections and accelerate the convergence speed. In contrast, our approach leverages past planning experience and can further reduce the likelihood of redundant sampling.

In light of the above, how to use past experience to speed up the solution of motion planning problems has attracted a lot of attention in the robot motion planning community. Roadmap-based approaches construct a graph data structure in configuration space to store connectivity information that has been collected in previous planning queries, while in the repetitive manipulation tasks of the robot arm, the position of the obstacle changes in workspace. An improved version of the original PRM algorithm, LazyPRM [13], was proposed to adapt to the changing environment by delaying collision detection. Rosell et al. collected demonstration data as a priori knowledge and used Principal Component Analysis (PCA) to learn the demonstration data to improve the path planning efficiency of the robotic arm during the grasping operation [14]. Similarly, PCA is used to learn demonstration data and combined with the RRT algorithm to achieve fast biased sampling [15], but this paper has only validated this in a 2D environment. Roveda et al. proposed an algorithm called HMM that is able to select reliable task trajectories from demonstration data [16], but when the workspace changes, some of the trajectories might conflict with the environment and thus planning fails. The road map constructed by GMM mixture distribution can quickly plan the path of grasping objects in a narrow passage [17]. Furthermore, the adaptive sampling method is combined with LazyPRM to solve the continuous operation motion planning of the manipulator. The past trajectories can be stored in the path library, which invoke a suitable path to solve the new motion planning queries. In the Lightning framework [18], the authors selected previous paths in a probabilistic way and the RRTConnect algorithm to modulate it. Similarly, the authors of [19] used previous experience to create a sparse roadmap, and then found valid path by A* [20], a shortest path search algorithm. On this basis, this method adopts a guaranteed mechanism so that when the A* search algorithm cannot find a feasible path, the RRTConnect algorithm can be used to repair from the disconnected states. However, these path library-based methods occupy a large amount of storage space, and in a semi-structured environment, some of the stored paths will fail due to changes.

Learning-based approaches are often used to analyze the potential characteristics of a task from a set of task-related expert demonstration trajectories by deep reinforcement learning [21], which are used to guide the generation of task-related motion plans. The conditional variational auto-encoder, a generative model, allows for coding the environmental features and sampling from the latent space in complex environments [22]. Motion planning networks [23,24] are a novel class of planners that attempt to solve motion planning problems by deep learning. These methods first encode the environmental point cloud data and then exploit neural networks to fit the expert’s demonstration trajectories, but these methods depend on the quality of the dataset and are subject to error accumulation when the neural networks are forward-propagated to generate samples. Reinforcement learning approaches treat the motion planning problem as a Markov process [25], where the intelligence learns the planning strategy through continuous trial and error, yet the manipulation skills learned by the robot in the simulation environment are difficult to deploy on real robots. Deep reinforcement learning-based planning algorithms have lots of model parameters and it is difficult to deploy on robotic arms.

This paper proposes a new motion planning algorithm, the lazy demonstration graph (LDG), which solves the repetitive motion planning of a manipulator. First, the high-quality trajectory solved by the expert planner is used as a priori knowledge to train the GMM, a generative model in machine learning, and the trained distribution is used to realize adaptive sampling so as to sample in the specific area containing the optimal paths, planning speed, and path quality. Second, this sampling method is combined with the LazyPRM algorithm. The introduction of lazy collision detection does not need to evaluate the overall validity of the graph to accelerate the construction of a roadmap. More importantly, it can adapt to environmental changes by querying candidate paths. Finally, we analyzed and modeled the continuous operation task of the manipulator, designed two different manipulator operation scenarios in the simulation environment, and achieved the continuous motion planning of the manipulator for the pick-and-place tasks by reusing the constructed LDG, without planning from scratch. In summary, the contributions of this paper are as follows:

1. In this paper, we propose an experience guided sampling method. The advantage of this method is that the distribution of demonstration data is learned via GMM, which are used to generate samples at the task-related location, so as to improve the sampling efficiency.

2. A new algorithm called Lazy Demonstration Graph (LDG), combined the above-mentioned sampling method with LazyPRM algorithm is proposed. The advantages of new algorithm are: First, the multi query property of graph is used to solve the continuous motion planning problem of the manipulator. Second, the introduction of lazy collision detection can improve planning efficiency and allows for adaptation to changes in the workspace to some extent.

3. The repetitive pick-and-place tasks are modeled, and the continuous planning problem of the robotic arm can be solved effectively by the proposed method and have been verified in the simulation experiments, at the same time, a physical experiment was designed to verify the effectiveness of our method.

The rest of this paper is organized as follows. Section 2 introduces the design process of our method in detail. Section 3 shows the results of the experiment. Section 4 provides a summary and future research plans.

## 2. Materials and Methods

### 2.1. Problem Definition

In this section, we focus on the definition of repetitive motion planning problems for robotic arms in semi-structured scenarios. Consider a *d* Degrees Of Freedom (DOF) robot, where the configuration space (C-space) is the *d*-dimensional space consisting of all possible configurations of the robot, and each possible configuration is a point in C-space, which represents the *d* joint angles of the robot.

Let $X\subseteq R^{d}$ be the C-space. We assume that $X_{obs}$ is the region occupied by obstacles in C-space; therefore, $X_{free}=X\backslash X_{obs}$ represents the collision-free region. Let $x_{start}$ denote the initial configuration in the collision-free C-space, and $x_{goal}$ is the target configuration. A query for motion planning is to find a continuous trajectory $\tau :\left[0,1\right]\to X_{free}$ for a given $x_{start}$, $x_{goal}$ such that each point of the trajectory lies in the collision-free region, where $\tau \left(0\right)=x_{start}$, $\tau \left(1\right)=x_{goal}$ and they represent the start and end of the trajectory, respectively.

Robotic arms are usually required to move objects to specific locations in the workspace, for example, shelf stacking (see Figure 1). We assume that an instance of the robotic arm manipulation task *T* is represented by a set $T=\{t_{1},t_{2},\ldots ,t_{i}\}$, where $t_{i}\left(o_{i},p_{start},p_{goal}\right)$ is one of the subtasks, which consists of the manipulated object $o_{i}$, start pose of the robot arm end-effector $p_{start}$ and target pose $p_{goal}$. Let function $f_{x}$ denote the mapping relationship from the pose of robot arm end-effector to a joint configuration of the robotic arm such that
(1)$$\begin{array}{cc} f_{x}\left(p_{start}\right) & =x_{start}\\ f_{x}\left(p_{goal}\right) & =x_{goal} \end{array}$$
where $f_{x}$ can be derived from the inverse kinematics of the robot. On this basis, each subtask $t_{i}\left(o_{i},x_{start},p_{goal}\right)$ corresponds to a new motion planning query that leads to changes in the objects’ position and in the underlying C-space of the robot indirectly. Furthermore, the repetitive manipulation task *T* is equivalent to multiple queries of motion planning for a finite time in semi-structured scenarios.

### 2.2. Learning Sampling Distribution

One of the main ideas of the LDG is to introduce the previously solved path information into the sampling distribution to effectively solve similar task instances. This distribution focuses on placing samples in parts of the configuration space that are more relevant to the task to reduce useless search. Limiting the sampling space not only improves the planning efficiency of the planner but also overcomes the problem that it is difficult to place samples in a complex environment.

Owing to the complex configuration space topology of the robotic arm, we use a Gaussian Mixture Model (GMM) to model the distribution of collision-free trajectories in the robot C-space, using a probabilistic approach to extract the key configurations of the demonstration trajectories. We iteratively estimate the parameters of the GMM by an expectation maximization (EM) algorithm [26] that maximizes the likelihood that past solution configurations are sampled from this distribution [27]. To speed up the training process as well as avoid local optimization, we initialize the parameters of the EM algorithm using the bisecting K-Means [28] clustering algorithm, while the k values are selected by the Bayesian Information Criterion (BIC) [29]. Sampling from the trained distribution can then create samples at task-relevant locations.

We use expert’s demonstration trajectories as priori experience. These high-quality solutions can be obtained by asymptotically optimal motion planners for specific operation tasks. We discretize these trajectories to obtain the key configurations, which contain most of the information of the previous solutions, and these key configurations constitute the training set *q*:(2)$$q={\left[q_{1}\;\ldots \;q_{i}\;\ldots \;q_{n}\right]}^{T}$$
where $q_{i}$ represents the path points contained in the *i*th demonstration path. We use this dataset to train a GMM, which can be considered as a superposition of multiple Gaussian models:(3)$$p\left(q\right)=\sum_{k=1}^{K}\pi_{k}N\left(q;\mu_{k},\sigma_{k}\right)$$
where p(q) represents the probability density function, *K* is the number of Gaussian components, and each Gaussian distribution *N* is composed of the mean $\mu_{k}$ and covariance matrix $\sigma_{k}$, and $\pi_{k}$ is the weight of the Gaussian distribution.

Parameter estimation for the Gaussian mixture distribution is more complex than that for the Gaussian distribution. Thus, we employ the expectation and maximization steps to make the distribution fit the data in the dataset. In the expectation step, we introduce the hidden variable $\gamma_{i,k}$, which represents the probability that the *i*th data come from the *k*th Gaussian component.
(4)$$\gamma_{i,k}=\frac{\pi_{k}N\left(q_{i};\mu_{k},\sigma_{k}\right)}{\sum_{k=1}^{K}\pi_{k}N\left(q_{i};\mu_{k},\sigma_{k}\right)}$$

In the maximization step, we use the updated probability value $\gamma_{i,k}$ to update the parameters of the GMM. First, we calculate the new mean of each Gaussian component:(5)$$\mu_{k}=\frac{\sum_{i=1}^{N}\gamma_{i,k}q_{i}}{\sum_{i=1}^{N}\gamma_{i,k}}$$

With the updated means, we can update the covariance matrix for each component:(6)$$\sigma_{k}=\frac{\sum_{i=1}^{N}\gamma_{i,k}\left(q_{i}-\mu_{k}\right){\left(q_{i}-\mu_{k}\right)}^{T}}{\sum_{i=1}^{N}\gamma_{i,k}}$$

Finally, new weights can be calculated for each component:(7)$$\pi_{k}=\frac{\sum_{i=1}^{N}\gamma_{i,k}}{N}$$

We terminate the iteration when the log-likelihood function reaches a local optimum:(8)$$\ln p\left(Q\right)=\sum_{i=1}^{N}\ln(\sum_{k=1}^{K}\gamma_{i,k}N_{k}\left(q_{i};\eta_{k},\sigma_{k}\right))$$

Once the model converges, we can use the distribution to generate samples, and the complete process is shown in Figure 2.

### 2.3. Lazy Demonstration Graph

In the previous section, we characterized the prior knowledge obtained by expert demonstrations with a GMM, and in this section, we generate sampling points and construct a probabilistic roadmap by Gaussian mixture distributions; that is, we apply previous solutions to solve specific motion planning problems. We first describe how the sampling-based motion planning algorithm constructs the roadmap, taking LazyPRM as an example, which is a classical roadmap-based motion planning algorithm on which our method is based.

As shown in Algorithm 1, the LazyPRM algorithm can be divided into two phases: constructing graph G=(V,E)⊆X and querying. In the graph construction phase, LazyPRM iteratively samples a random configuration $x_{rand}$ without checking its validity by collision detection. This random configuration is added directly to the set of vertices *V* (line 4 and 5). For each sample $x_{rand}$, a nearest neighbor search is performed (line 6–10); in this step, we usually have to set the value γ for distance constraint between two configurations. If the configuration *x* in the set of vertices *V* and random configuration $x_{rand}$ meet the distance constraint ($||x-x_{rand}\left|\right|<\gamma$), then the configuration *x* is considered as the nearest neighbor $x_{near}$ of the random configuration $x_{rand}$. The random configuration $x_{rand}$ is also directly connected to its nearest neighbor $x_{near}$ as edge in the graph without collision detection, and this step often called local planning (line 9). When the graph is constructed, the shortest path can be found by the graph-search-based algorithm A* or Dijkstra’s algorithm (line 13). Because the lazy data structure allows for invalid edges, the path obtained by the search algorithm may collide with obstacles; therefore, collision detection is needed for the solved path. If a collision occurs, the candidate path is searched until a collision-free effective path is obtained (line 14).
**Algorithm 1:** Lazy PRM
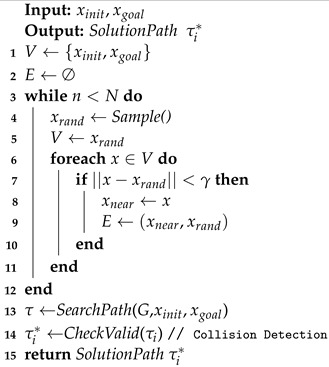


In the offline phase, we first collect the high-quality trajectories related to the operation task, then discrete the trajectories to form the training set, and train the GMM until convergence, thus the distribution GMM_Sample() can be used for sampling in task-related areas. The overall flow of the LDG is shown in Algorithm 2. The algorithm is divided into two phases, as in most roadmap-based algorithms. First, we construct an undirected graph structure through cyclic sampling, and then use it to query collision-free paths. The specific operation task *T*, the number of initial samples *N* and the distribution GMM_Sample() are used as inputs to the algorithm. We first initialize the node set *V* and edge set *E* of the graph *G*. Consistent with what was performed in these papers, in order to balance exploration with exploitation, we set a threshold value *t* (line 3) to choose whether to use a Gaussian mixture distribution or uniform distribution to generate samples in a probabilistic way. Lines 4–10 represent the cyclic sampling phase, and when the random number *k* lies between 0 and *t*, we use the trained Gaussian mixture distribution to generate a sample by first selecting a Gaussian component and then drawing samples from this Gaussian distribution, with each Gaussian component being selected with a probability proportional to its probability of being in the mixture distribution. Accordingly, the samples are placed in a promising location, reducing useless exploration; otherwise, the sample is generated through a uniform distribution to achieve the purpose of exploring the C-space. Because we introduce the lazy collision detection strategy, we do not make a judgment here on whether the sample $x_{rand}$ collides with the environment, but add it directly to the node set V (line 11). As in Algorithm 1, we use nearest the neighbor search to obtain the nearest neighbor node $x_{near}$ of $x_{rand}$ and construct connections between $\left(x_{near},x_{rand}\right)$ via a local planner, which can connect two nodes in a straight line to form an edge in the graph in lines 12–17.

Once the undirected graph data structure *G* is constructed, multiple queries can be performed by the persistent graph structure to solve continuous motion planning problems, as shown in Figure 3. The received task is processed in lines 20–24, and each subtask $t_{i}$ corresponds to a request of motion planning. The function SearchPath(·) is used to process each planning request, and it contains three parameters. Here, $p_{start}$ is the current pose of the end effort, $p_{goal}$ is the target pose of the end effort, and function $f_{x}\left(\cdot \right)$ represents the mapping relationship from the end effort of robotic arm to the joint configuration, which is used to determine the joint configuration of the arm for object manipulation, the function $f_{x}\left(\cdot \right)$ can in turn be deduced according to the inverse kinematics of the manipulator. In this way, the path search function takes a pair of robotic arm configurations as its input and searches for the shortest collision-free path by the graph *G*, with the help of priori knowledge. The graph search algorithm we use is the A* algorithm based on depth-first search.
**Algorithm 2:** Lazy Demonstration Graph
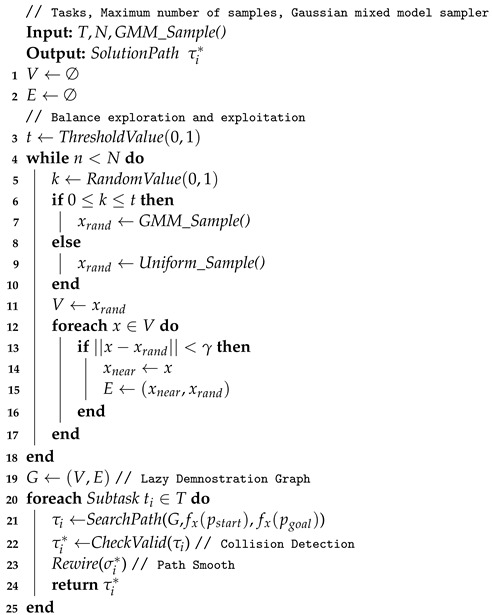


Some valid nodes and edges of the graph constructed in C-space are invalidated because the manipulated object is attached to the manipulator. This results in the possibility of a collision between the manipulator and the environment increasing, as well as the position of object changing due to the operation of the robot. Therefore, in the construction phase of the graph, judging the validity of the nodes and edges of the graph is unnecessary, but collision detection on the searched paths must be performed $\sigma_{i}$ through the function CheckValid(·) (line 21). The collision detection function is implemented by the Flexible Collision Library (FCL); FCL is an open source project that includes various techniques for efficient collision detection and proximity computation [30]. If there is a collision on the path, the colliding nodes and edges are moved out of consideration, and then the candidate path $\sigma_{i}^{*}$ is obtained by graph search. In this way, a lot of computation time can be saved without evaluating the overall effectiveness of nodes and edges in the graph, and we can also ensure the normal progress of subsequent tasks. Because the local planner connects the two nodes in a straight line, it leads to the redundancy of path points. Thus, a function Rewire(·)—is introduced to remove redundant nodes in the path by evaluating whether the linear trajectories of two discontinuous nodes in the connection path have no collision (line 22). Similar to the reselection parent node algorithm used in RRT*, the lightweight implementation of this algorithm has little processing overhead, so it can be used without significantly increasing the path generation time, as shown in Figure 4.

## 3. Experiments

To evaluate the effectiveness of our method, we designed a three-part experiment. First, a path planning test was conducted on a 2D grid map with a translational-only point robot, which illustrates the creation and uses of the LDG in a more visual way. Subsequent experiments were deployed on the ROS and MoveIt! In the simulation environment, two scenes of shelf stacking and picking from bin were built. Continuous pick-and-place manipulation experiments with a 6 DOF AUBO−i5 robotic arm showed that our method had a certain degree of efficiency and robustness. Finally, we use MoveIt! in ROS to plan the path of the physical manipulator, so as to realize the task of picking and placing. All tests were carried out on an Intel Core i5 with 2.40-GHz cores and 16 GB of RAM running Ubuntu 16.04.

### 3.1. 2D Gridworld

In this part of the experiment, we compared our method with the LazyPRM, RRT*, RRTConnect [31], and BiRRT* [32] algorithms. The LazyPRM algorithm constructs the road map with uniform sampling. The RRTConnect algorithm is the greedy variant of RRT, which is rapidly extended by two trees, and has great speed advantage in solving motion planning problems. BiRRT* is the state-of-the-art single-query algorithm, which inherits the asymptotic optimization property of the RRT* algorithm and has a faster expansion speed. Our method sets the same maximum number of samples to 470 as in LazyPRM, and the connection distance of the nearest neighbor node is set to 45. We used BiRRT* as our expert tutorial planner to tutorialize the planning tasks and used the discrete path points as training data for the GMM. The number of iterations of the EM algorithm is set to 100, and the optimal number of components of the GMM is 12 according to the BIC criterion. probability of generating samples using GMM was set to 0.3. All code of this part of the experiment was implemented in matlab2017.

Figure 5 shows the planning visualization of our proposed method compared with three representative methods, i.e., LazyPRM, RRTConnect, and BiRRT*, in two different 2D scenes. Each 2D scene has a length and width of 500, and consists of several gray squares for obstacles. The planner needs to find collision-free paths given the start and end points within the limit time.

We evaluated the performance of the algorithms from the three aspects of planning time, path cost, and success rate. Planning time measures the time spent by the planner to generate a path from the starting point to the goal point. Path cost reflects the ability of the planner to approach the optimal path. When all the states in the path are located outside the obstacles and the planner is not timed out, then the path is considered successful. Then, 30 tests were run in two scenes, and the experimental results were averaged; all the results of the four algorithms are shown in Table 1 and Table 2.

As can be seen in Table 1 and Table 2, LDG undergoes a greater improvement in the quality of the paths and success rate of planning compared to the LazyPRM algorithm, in the two scenarios, the length of the path decreases by 12.53% and 9.8%, respectively, and the planning success rate improved by 36.67% and 53.34%, so the efficiency has been improved. Quantitatively, the planning time of RRTConnect algorithm is slightly better than that of the LDG, but the path quality of our algorithm is improved by 28.9% and 36.9%, respectively, compared to RRTConnect. Compared with the BiRRT* algorithms, the LDG is slightly better than these two algorithms in terms of path quality, and has a significant advantage in planning time, the planning time in each of the two scenarios was reduced by 85.3% and 88.7%. The above three evaluation indicators are better than RRT*. We also introduce the path smoothing strategy, which enables the quality of the path to be further improved and also guarantees that the quality of the path is good even with less prior knowledge.

Overall, the performance of our algorithm is the best compared to the other four. In contrast, the LDG, such as the LazyPRM algorithm, delays the collision detection to the search phase of the path, which not only improves the speed of graph construction but also lets the LDG adapt to changes in the workspace to some degree. When the environment has partially changed, the LDG can also search the candidate path through the constructed graph. More importantly, for new planning problems, the tree-based method needs to explore the robotic arm configuration space again each time. The LDG algorithm, as a raodmap-based method, can save a large amount of information in the robotic arm configuration space in advance. In subsequent planning tasks, only directly calling the graph structure and performing the search for paths is needed. This approach also improves the planning efficiency. In the subsequent experiments, we used the multi-query nature of the LDG algorithm for continuous path planning for operation-oriented tasks.

### 3.2. Continuous Pick-and-Place of Manipulator

In this section, we design a robotic arm continuous motion planning experiment for an operational task in the Robot Operating System (ROS) environment. We first describe the experimental environment and experimental setup, and then compare the LDG with other robust planners in the same environment. The simulation environments included shelf stacking and picking from a deep bin, which are shown in Figure 6a,b separately. The robotic arm used is the AUBO−i5 with an attached Robotiq gripper. Taking the shelf stacking task as an example, this task required the robot arm to pick up all cylinders on the left side of the shelf and place them on its right side while avoiding obstacles in the workspace during the robot arm planning process. The operation task consists of four grasping and placing subtasks, for a total of eight motion paths.

We tested the LDG with the RRTConnect and BiRRT* algorithms in the same scenario and performed 10 iterations of the experiment for the shelf stacking task, with a total of 80 paths (picking and placing 40 times). As in the previous 2D experiments, we first used the BiRRT* algorithm to teach the operational task, and then used these demonstration paths as training data to train the GMM until convergence, the number of iterations of the EM algorithm is set to 100, the number of Gaussian components is 14 according to the BIC criterion, this step was performed offline. In practice, we do not teach all the picking and placing tasks by BiRRT*, because the paths planned in the same environment tend to have similar characteristics, a small number of paths are already representative. We also provide a certain percentage of uniform samples, which further improves the generalization ability of the planner. With the help of a trained Gaussian mixture distribution, the LDG can place more samples at task-relevant locations. The trajectory smoothing strategy was deployed for each planner, which could effectively improve the quality of the paths. All algorithms in this part of the experiment were implemented by C++ codes.

The experimental results are shown in Table 3 and Table 4. Each subtask contained one picking and placing manipulation, and we measured the performance of the planner planning mainly from the two metrics of path cost and planning time. In the shelf stacking task and picking from a deep bin task, the initial number of nodes of the LDG was set to 3000 and 4500, and the ratio of Gaussian mixed sampling to uniform sampling was 1:2. We show the experimental data for the first two subtasks in the table, which shows the effectiveness of our proposed method, and the overall performance of the three planners in the four subtasks is compared more visually in box plots.

As can be seen from the data in Table 3 and Table 4, the LDG has a higher time cost than RRTConnect for the first picking operation, as the roadmap-based motion planning algorithm needs to construct a graph and save it. However, in the subsequent planning process, the LDG can reuse the saved graph structure in subsequent subtasks to achieve the effect of increasing the solution speed. Because RRTConnect and BiRRT* are tree-based motion planning algorithms, each query requires expanding the entire tree from scratch, which does not save computation time. Furthermore, because of the introduction of prior knowledge, the LDG can generate a large number of samples directly at these narrow passages in the workspace, so the average planning time of the LDG is smaller than the other two algorithms. In summary, when performing two picking and placing tasks, our method reduces the planning time by 39.1% and 34.1% compared to the RRTConnect algorithm in the two experimental scenarios, and the length of the path is reduced by 53.2% and 38.5%, respectively. Compared to the BiRRT* algorithm, our method reduces the planning time by 80% and 76.8% in the two experimental scenarios, and the length of the path is reduced by 20.9% and 15.5%, respectively.

Figure 7 visually shows the performance comparison of the three planning algorithms used by the AUBO−i5 manipulator to perform shelf stacking tasks in the form of box plots. The top plot corresponds to the planning time, and the bottom plot corresponds to the length of the path. Task 1 and Task 4 are shown from left to right, and each task contains a pick-and-place operation, represented by blue and orange boxes, respectively. It can be seen from the box plot that the LDG has the best performance in general, except the time spent in the pick operation of task 1 is higher than that of RRTConnect. Second, the algorithm has fewer outliers, which also reflects that the LDG algorithm has better robustness. These experimental results show that the LDG constructs a lazy probabilistic road map by introducing prior knowledge, which is suitable for repetitive operation motion planning in similar environments.

Figure 8 shows composite pictures of AUBO−i5 performing a shelf stacking task. As can be seen from task 1, the trajectory generated by the LDG algorithm is close to the optimal trajectory. When the pick-and-place manipulation has completed, we added a new obstacle, which invalidated some edges in the constructed roadmap, and the LDG was able to complete subsequent manipulation by querying candidate paths. This approach ensured the stability of the motion planning process, which benefits from the lazy data structure of the LDG in each task.

### 3.3. Physical Robot Arm Planning Experiment

In order to further verify the effectiveness of the method proposed in this paper, we use the AUBO−i5 physical manipulator to build a scene for picking and placing tasks, and use MoveIt! in ROS, the software platform builds a corresponding simulation environment according to the real-world scene, as is shown in Figure 9a,b.

In this section, we use ROS as the host computer to control the physical manipulator, so that the AUBO−i5 manipulator can pick objects from box1 and place them into box2, while avoiding obstacles in the workspace. We first used the BiRRT* algorithm to demonstrate the first pick-and-place task, and used the demonstration data as prior knowledge to construct an adaptive sampler by GMM. Then perform the second picking and placing operation, and use the current position as the starting point for the motion planning problem Figure 10, we use the proposed method to plan the path of the robotic arm, so that its gripper moves to box1 to grasp the object; the path sequence is shown in Figure 11.

Similarly, we use the current position as the starting point for the placement task Figure 12, and to verify that our method can cope with changes in the workspace, we added a new obstacle between the two boxes, it can be seen that AUBO−i5 can still plan to the target position while avoiding obstacles in the workspace, realizing the placement of object; the path sequence is shown in Figure 13.

It can be seen from the experiments that our algorithm can be transferred from the simulation environment to the physical manipulator, and can better complete the motion planning for grasping and placing tasks, and at the same time, it can cope with changes in the workspace to a certain extent. The joint position change curve is shown in the following Figure 14 and Figure 15.

## 4. Conclusions

In this paper, we proposed LDG, a new motion planner for the pick-and-place tasks of a robotic arm. This method is a roadmap-based motion planning algorithm that uses the solutions of previous problems to speed up the solution of similar problems. We first analyzed and modeled the operation task-oriented planning problem of robotic arm, and then collected the high-quality trajectories planned by the expert planner as a priori knowledge. Notably, GMM fit an arbitrary distribution, therefore being able to learn the distribution of collected trajectory samples and build an adaptive sampler. Thus, GMM achieved sampling in specific regions of the optimal path solution to improve the speed and quality of motion planning. At the same time, we integrated this adaptive sampler into the LazyPRM algorithm, which enabled LDG to adapt to environmental changes as in a tree-based planning algorithm by delaying collision detection and guaranteeing the valid execution of subsequent tasks. For the new planning problem, the tree-based motion planning algorithm needed to plan from scratch, while we took advantage of the multi-query nature of the graph by saving the constructed road map at the end of the first planning task. In the follow-up planning problem, the persistent graph structure was called directly for the subsequent problem, thus further improving the speed of planning. We applied the proposed method to the continuous pick-and-place task of the AUBO-i5 manipulator, and both the simulation experiment and the physical experiment verified the effectiveness of our method. In the future, we will explore the combination of task planning with our proposed motion planning algorithm to solve complex long-term robot planning problems.

## Figures and Tables

**Figure 1 biomimetics-07-00210-f001:**
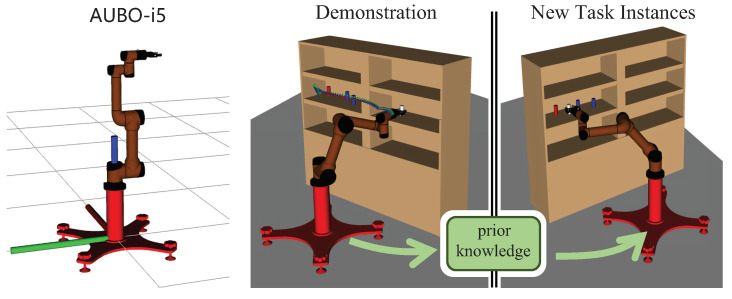
Our planner can generalize and leverage the high-quality paths solved by the expert planner to solve new task instances. The expert planner plans the paths by searching the manipulation tasks and collecting and using these paths as priori knowledge to solve new motion planning problems quickly and consistently to grab all the cylinders on the left and place them on the right side of the shelf, thus avoiding collisions.

**Figure 2 biomimetics-07-00210-f002:**
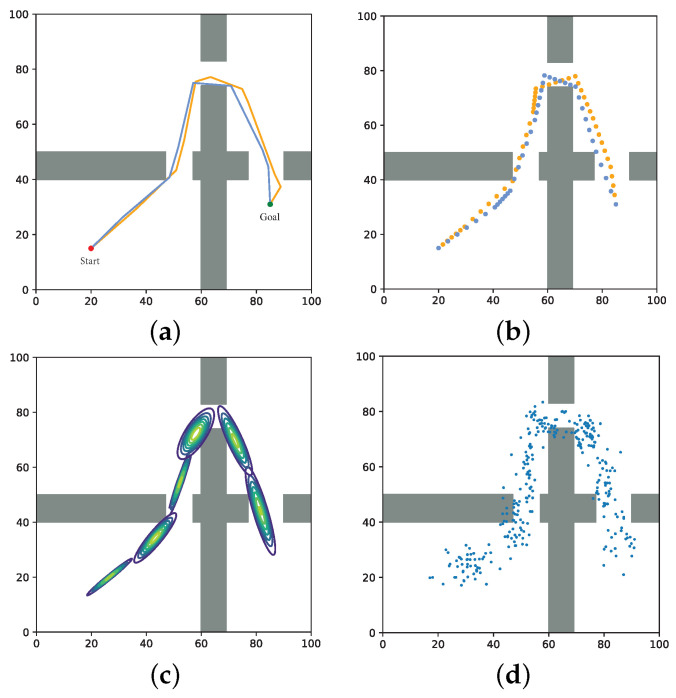
Overall process of the learning sample distribution. Collecting successful and high-quality planning cases (**a**), discretizing the trajectory, and constructing the training set (**b**), training the GMM with the collected data until the model converges (**c**), using the trained model for generating samples in task-related locations (**d**).

**Figure 3 biomimetics-07-00210-f003:**
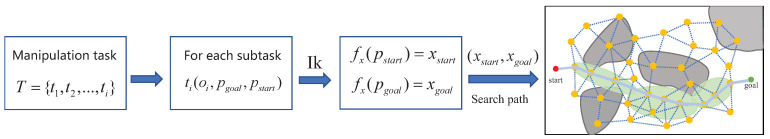
Overall process of solving the repetitive motion planning problem of robotic arm for manipulation tasks using LDG. The manipulation task *T* contains multiple pick-and-place subtasks and for each subtask $t_{i}$, the robotic arm is required to move the object from one position to another. the starting and target configurations of the arm can be obtained by inverse kinematics (IK) and then the shortest path is searched using the constructed LDG. the lazy data structure of LDG is kept in each task, which allows invalidating edges, it helps to improve the composition speed on the one hand and adapt the changes of the environment. In this figure, the green ellipse represents Gaussian mixture distribution obtained from prior knowledge training, which is used to place the samples in more promising locations.

**Figure 4 biomimetics-07-00210-f004:**
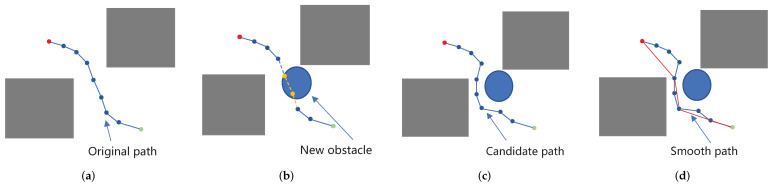
Given the start and goal points (shown in green and red dot), a path can be found through the graph search algorithm in a two-dimensional (2D) environment (**a**). When a new obstacle (blue ellipse) appears in the environment, the nodes and edges of the constructed graph structure will partially invalid (shown in yellow) (**b**). The planning problem is completed by searching the candidate path (ignoring the invalid nodes and edges when querying and searching path using other nodes and edges in the graph) (**c**). The path is further optimized by removing redundant states from the path (shown in red line) (**d**).

**Figure 5 biomimetics-07-00210-f005:**
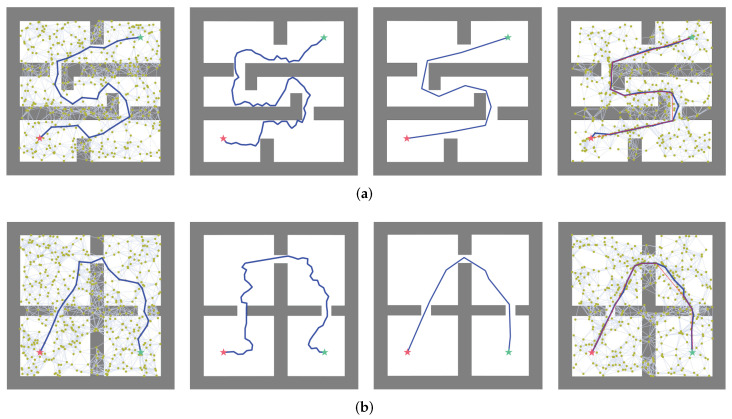
(**a**,**b**) are two different environments used in the 2D Gridworld experiments. From left to right, the planning results of LazyPRM, RRTConnect, BiRRT*, and LDG are shown. The green and red five-pointed stars represent the start and the goal, respectively; the blue line is the valid path generated by the planner; and the red line is the path of the LDG smoothed by the Rewire function, which can reduce the redundant state of the path.

**Figure 6 biomimetics-07-00210-f006:**
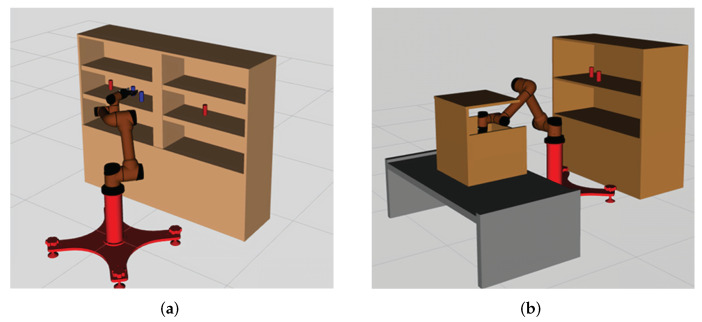
AUBO−i5 performs continuous pick-and-place tasks, shelf stacking (**a**), picking from deep bin (**b**).

**Figure 7 biomimetics-07-00210-f007:**
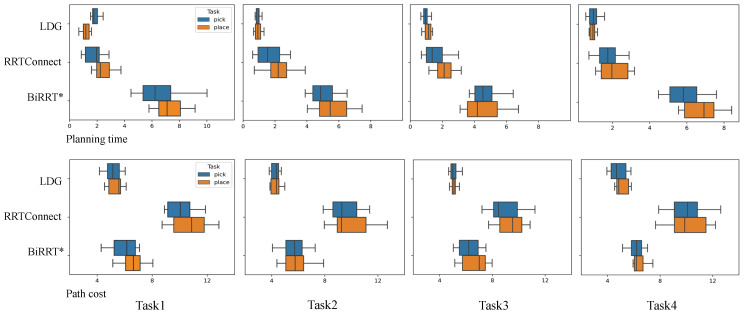
Experimental data obtained from the AUBO−i5 shelf stack tasks, which includes 4 times pick-and-place subtasks. Blue boxes and orange boxes are for pick-and-place tasks, respectively. To complete the manipulation task, each planner must plan a collision-free path to pick up the object and place it at the target location. The top boxplots represent the planning time, and the bottom boxplots represent the length of the planned path.

**Figure 8 biomimetics-07-00210-f008:**
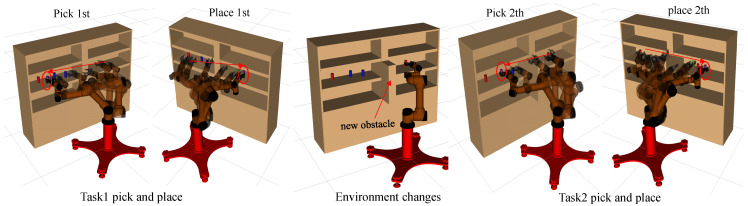
The composite images of the AUBO−i5 in shelf stacking task. These pictures show the robot executing a series of joint commands generated by LDG, picking up objects from the shelf and placing them on the other side. Although the roadmap in LDG algorithm is built for specific scenes, when the environment changes (the position of the operated object changes or new obstacles appear), the algorithm can still ensure the completion of subsequent tasks by querying candidate paths.

**Figure 9 biomimetics-07-00210-f009:**
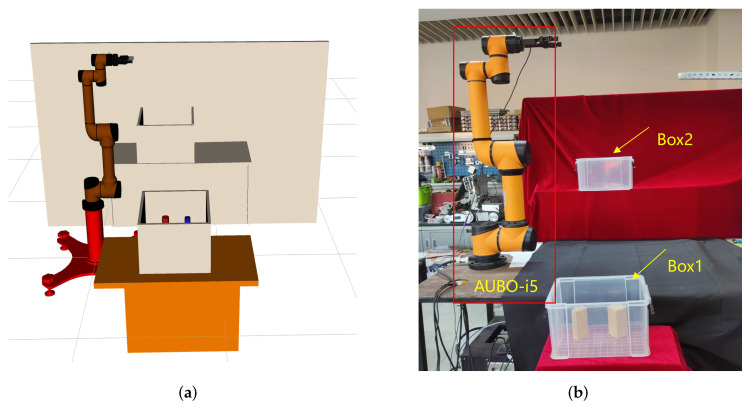
The scene for pick-and-place tasks in MoveIt! (**a**), AUBO−i5 physical manipulator (**b**).

**Figure 10 biomimetics-07-00210-f010:**
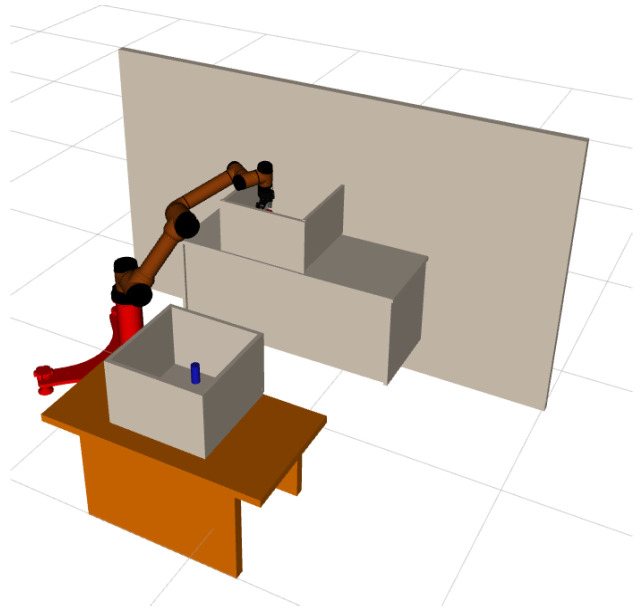
The starting position of AUBO−i5 when it is about to perform picking task.

**Figure 11 biomimetics-07-00210-f011:**
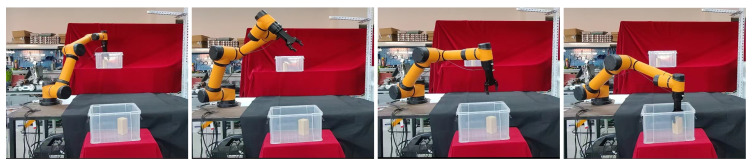
The path sequence of AUBO−i5 when executing the picking task.

**Figure 12 biomimetics-07-00210-f012:**
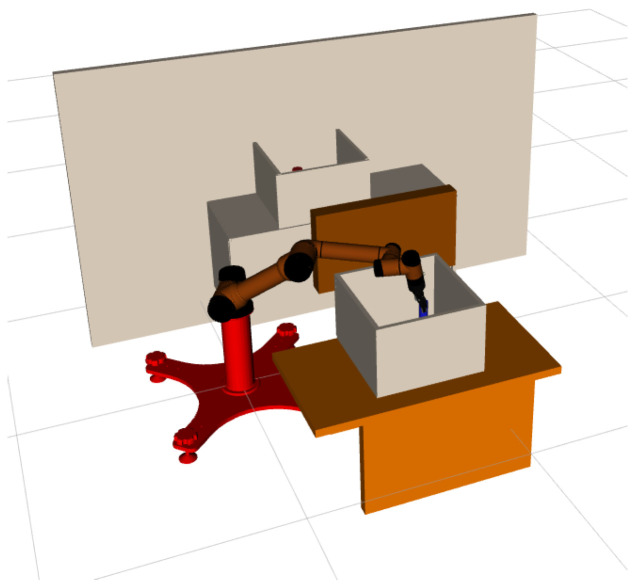
The starting position of AUBO−i5 when it is about to perform picking task.

**Figure 13 biomimetics-07-00210-f013:**
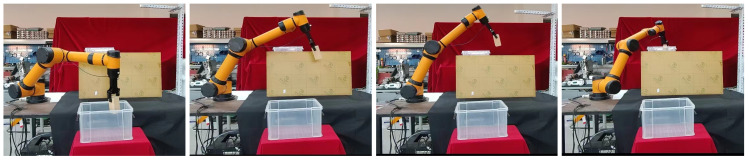
The path sequence of AUBO−i5 when it is about to perform placing task.

**Figure 14 biomimetics-07-00210-f014:**
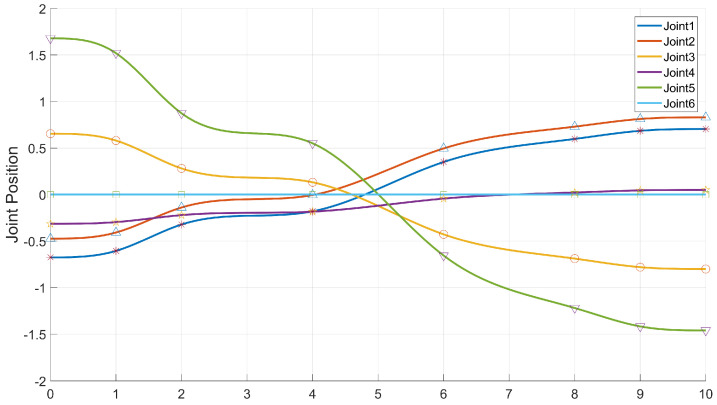
The joint position change curve of the AUBO−i5 when performing the picking task.

**Figure 15 biomimetics-07-00210-f015:**
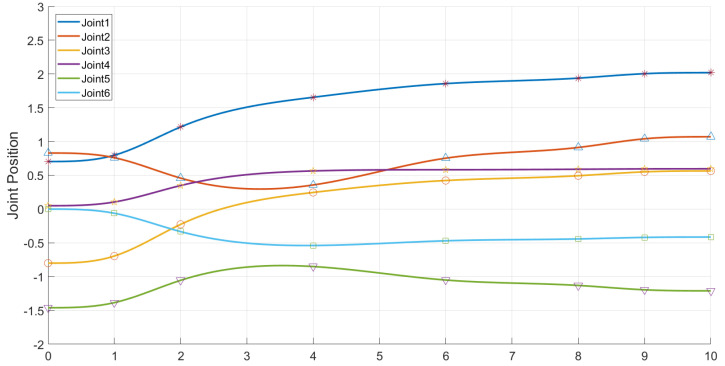
The joint position change curve of the AUBO−i5 when performing the placement task.

**Table 1 biomimetics-07-00210-t001:** Speed, path cost and success rate of LDG benchmarked against other algorithms in scene 1.

Algorithms	Planning Time	Path Cost	Success Rate
LazyPRM	0.376	975.63	63.33
RRT-Connect	0.229	1200.25	100
RRT*	3.716	890.27	83.33
BiRRT*	2.637	869.33	100
LDG	0.387	853.41	100

**Table 2 biomimetics-07-00210-t002:** Speed, path cost and success rate of LDG benchmarked against other algorithms in scene 2.

Algorithms	Planning Time	Path Cost	Success Rate
LazyPRM	0.295	726.21	46.66
RRT-Connect	0.286	904.81	100
RRT*	4.841	678.47	76.66
BiRRT*	2.811	663.39	100
LDG	0.319	654.88	100

**Table 3 biomimetics-07-00210-t003:** Numerical results of AUBO−i5 repetitive motion planning in shelf stacking task.

Algorithms	Tasks	Sequence	Planning Time	Path Cost
LDG	pick	1st	1.86	5.16
		2nd	0.95	4.26
	place	1st	1.19	5.34
		2nd	0.93	4.32
RRT-Connect	pick	1st	1.76	10.14
		2nd	1.63	9.49
	place	1st	2.43	10.73
		2nd	2.27	10.01
BiRRT*	pick	1st	6.67	5.98
		2nd	5.04	5.73
	place	1st	7.28	6.6
		2nd	5.66	5.82

**Table 4 biomimetics-07-00210-t004:** Numerical results of AUBO−i5 repetitive motion planning in picking from bin task.

Algorithms	Tasks	Sequence	Planning Time	Path Cost
LDG	pick	1st	4.54	6.47
		2nd	2.14	6.76
	place	1st	2.07	6.69
		2nd	2.3	7.04
RRT-Connect	pick	1st	3.16	9.24
		2nd	3.82	10.66
	place	1st	4.06	11.31
		2nd	5.91	12.66
BiRRT*	pick	1st	9.96	7.57
		2nd	10.49	7.98
	place	1st	12.51	8.66
		2nd	14.69	7.69

## Data Availability

Not applicable.

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
