# Peer review of "An Efficient Motion Planning Method with a Lazy Demonstration Graph for Repetitive Pick-and-Place"

_biomimetics, 2022, doi:10.3390/biomimetics7040210_

Round 1

Reviewer 1 Report

This paper presents a robotic arm planning method using the demonstration experience as a priori knowledge to improve the efficiency and quality of path planning when performing pick-and-place tasks, the simulation results verified the effectiveness. Generally, the contents are interesting and the whole paper seems to be logically. The reviewer recommends this paper to be published in this journal after major revision. The following comments may help to improve this paper:

1. The description of the methodology in the abstract is not sufficient enough. GMM algorithm is applied in the method section, which is not mentioned in the abstract.

2. This paper makes use of the GMM algorithm and the learning from demonstration, but the state-of-art related to GMM and learning from demonstration are not analyzed, it is recommended to add related description to clarify the paper’s originality.

3. Some experimental parameter settings are not specified. GMM algorithm requires initialization of the parameters, which are not given in the experimental section. It is suggested to give a description in experimental part regarding the setting of GMM parameters, such as the specific number of Gaussian components, etc.

4. Collision detection is mentioned several times, but how it is implemented in the experimental part of the article is unclear.

5. Others: the format, English can be improved and some errors need to be checked carefully and revised. For example, in 8th page, the caption of figure 4, “After the LDG is constructed……”.

Reviewer 2 Report

The paper addresses an interesting topic in the field of sampling-based planning algorithms, by introducing modifications in the process of collecting data. There are, however, some points that need to be discussed in more detail.

In the introduction, it is stated that the configuration space will be “modified” based on the task assigned to the robot. Configuration space is an intrinsic character of a robotic system and cannot be affected based on the task assigned to the robot. But the configuration of the robot on its configuration space is subject to change (based on the achieved configuration). Perhaps the change in the work space is meant? This point should be clarified in detail.

It seems that the configuration space is considered as a synonym for joint space(?) The case study of the paper is stated to be the manipulators, and an open-chain robot is considered as the example. It is stated that the answer to the inverse kinematics problem will be considered as target configuration. Inverse kinematics, however, for such robots generally returns multiple answers. The following points need be discussed: How the configuration feasibility of the solution is guaranteed? How the solution, between multiple options, is chosen? How is it guaranteed, that there is no configuration changes happening?

The abbreviation “C-space” is more common in the literature than “C space”. Also “Work space” is more common than “Working space”.

Line 28: “natural of manipulation”: “nature of manipulation”?

A comparison between the already existing incremental planning algorithms, which utilize the previous knowledge in the planning process, should also be discussed in the introduction.

Line 119: The correct mathematical sign for set subtraction is \ and not /. Hence, X_{free}=X\X_{obs}.

Line 121: The target of a trajectory map should be the C_{free}, and not \mathbb{R}^d. It may be better, that not the same variable is used for trajectory and the state. (In Algorithm 1 the variable \tau is utilized for the path.)

In (3), variable q is not well-defined. Should it be q_i?

The captions of Figures 2(c) and 2(d) are missing.

In the caption of Figure 3: Is by “positional”, “position” (or “pose”) meant?

The functions “Sample()” and “FindNearstNeighbors(xrand)” of Algorithm 1 need to be defined.

Line 183: How is it possible that a graph, constructed via discrete samples from C-space, represents exactly the continuous C-space?

Line 189: Is the cross-reference “line 9-10” correct? It points to the end of loops.

Line 191: “Djkstra algorithm”: “Dijkstra’s Algorithm”?

Algorithm 2 is unfortunately not easily readable. The first three lines are not elaborated in detail. The input comes after some lines, and there is no output declared. Many functions are not defined. It is not clear, how the collision check is performed. A nomenclature improves readability.

Please make sure, that all lines start with a capital letter. See for example Line 205. The names and abbreviations are also not consistent. They are sometimes in lower, and sometimes in upper case.

Line 216: What is by “local planner” meant?

Line 217: What is “LDG G”?

Reviewer 3 Report

The paper proposes the lazy demonstration graph planner for the pick-and-place tasks of a robotic arm. The topics of the paper are interesting and suitable for the journal. The following points, however, need to be clarified to improve the quality and the readability of the manuscript.

(1) The title of the manuscript must be modified to highlight the innovation of the manuscript. And moreover, the novelty of this paper with respect to the state of art is not clearly described. It is not easy to assess which is the main contribution of the work with respect to the present literature.

(2) Note that the semistructured environments is only in the title of the paper, and thus, the author should explain its meaning. And moreover, how do the configurations and dynamics of the selected robots (such as the collision dynamics during pick-and-place tasks), as well as the semistructured environments affect its motion planning with lazy demonstration graph.  

(3) The disadvantages of the proposed lazy demonstration graph planner, which refer to “lazy”, should be investigated.  

(4) Only numerical experiments and results are presented in the manuscript. The results seem interesting but some experiments must be used to verify the obtained results.

Round 2

Reviewer 1 Report

 After checking the revised version, I think all my concerns have been well addressed. I have no further comments, the revised paper can be accepted as it is.

Author Response

The authors would like to express their sincere gratitude to the reviewer again for the constructive comments yet critical comments provided in helping us to improve the quality and presentation of the paper further.

Reviewer 3 Report

There has been a great improvement in the revised paper, the following problems, however, need to be further addressed. First, note that the semistructured environments is only in “ Introduction” of the revised paper, and in fact, it is the working condition of the investigated robot. And thus, the author must investigate that how do the semistructured environments affect the motion plan of the robots. Second, a fully restrained six-dof AUBO-i5 robotic arm, which is different from the less and redundant restrained robots in terms of dynamics, is employed to show that the efficiency and robustness of the proposed method. However, the configurations and dynamics, especially the collision dynamics during pick-and-place tasks, as far as I know, will have an impact on the motion planning of the robotic arm. The problems must be addressed to obtain the efficient motion planning and control for the robotic arm. Finally, only numerical simulations are presented with the ROS in the revised manuscript, and furthermore, some physical experiments must be used to verify the proposed method and the obtained simulation results. 
